# Long-Term Effects of Initiating Continuous Subcutaneous Insulin Infusion (CSII) and Continuous Glucose Monitoring (CGM) in People with Type 1 Diabetes and Unsatisfactory Diabetes Control

**DOI:** 10.3390/jcm8030394

**Published:** 2019-03-21

**Authors:** Jon-Duri Senn, Stefan Fischli, Lea Slahor, Susanne Schelbert, Christoph Henzen

**Affiliations:** Division of Endocrinology, Diabetology, and Clinical Nutrition, Department of Medicine, Lucerne Cantonal Hospital, CH-6000 Lucerne 16, Switzerland; jonduri.senn@unibas.ch (J.-D.S.); stefan.fischli@luks.ch (S.F.); lea.slahor@luks.ch (L.S.); susanne.schelbert@luks.ch (S.S.)

**Keywords:** Type 1 diabetes, continuous subcutaneous insulin infusion, insulin pump, functional insulin therapy, continuous glucose monitoring

## Abstract

Background: We aimed to assess the long-term effects of the introduction of continuous subcutaneous insulin infusion (CSII) and continuous glucose monitoring (CGM) in people with type 1 diabetes (T1D). Methods: A prospective single-centre cohort study including participants with T1D and HbA1c > 7.5%. After completing a course in flexible intensified insulin treatment (FIT), participants were offered treatment change to CSII/CGM. FIT participants with HbA1c ≤ 7.5% who remained on multiple daily injections (MDI) and without CGM were monitored as a separate cohort to compare the cumulative incidence of diabetic complications. Results: The study cohort included 41 participants with T1D (21 male/20 female). The mean age (±SD) at inclusion was 24.2 ± 10.9 years, the mean follow-up was 8.9 ± 2.8 years, and the mean diabetes duration at the end of the study was 15.9 ± 10.1 years. The mean HbA1c level before the introduction of CSII was 8.8 ± 1.3% (73 ± 8 mmol/mol), and decreased significantly thereafter to 8.0 ± 1.1% (63 ± 7 mmol/mol) (*p* = 0.0001), and further to 7.6 ± 1.1% (59 ± 11 mmol/mol) after the initiation of CGM (*p* = 0.051). In the MDI group the HbA1c levels did not change significantly during a mean follow-up of 6.8 ± 3.2 years. The frequency of severe hypoglycaemia after the introduction of CSII/CGM declined significantly (from 9.7 to 2.2 per 100 patient-years, *p* = 0.03), and the cumulative incidence of newly diagnosed diabetic microvascular complications were comparable between the study group and the observational cohort. Conclusion: In people with T1D and unsatisfactory diabetes control the introduction of CSII and CGM results in a substantial and long-term improvement.

## 1. Introduction

In patients with type 1 diabetes mellitus (T1D), the objectives of intensified insulin treatment regimens are twofold, i.e., to achieve strict glycemic control in order to minimize the risk of secondary diabetic complications, and to avoid severe hypoglycemia and ketoacidosis to attain the best possible quality of life [1,2,3].

Intensive insulin treatment aims at mimicking physiological insulin secretion patterns by means of multiple daily injections (MDI) of long-acting basal insulin and short-acting bolus insulin or via continuous subcutaneous insulin infusion (CSII) (insulin pump therapy) guided by frequent measurements of blood glucose either by self-monitoring (SMBG) or by continuous glucose monitoring systems (CGMS) [4]. Prerequisites for an effective insulin therapy are the knowledge of the individual amount of basal or fasting insulin, of the carbohydrate content of the food (“carbohydrate counting”), and the corresponding requirement of bolus insulin, as well as the proper insulin adjustments in defined situations, such as physical activity, sickness, or driving. Therefore, in addition to the correct way of insulin delivery, structured and regular diabetes education is a cornerstone of the successful management of diabetes mellitus [5].

There is some debate on whether CSII is superior to MDI with respect to HbA1c (glycated hemoglobin reflecting the average three-months plasma glucose concentration), hypoglycaemia reduction, and quality of life. A number of meta-analyses yielded only modest improvements in unselected patient populations (with a range of the mean difference of the HbA1c before/after studies of 0.21–0.72%) [6,7]. However, in people with T1D and frequent severe hypoglycemia or persistently high HbA1c > 9%, the switch from MDI to CSII proved to be much more beneficial. With regard to upcoming technology, such as the artificial pancreas (“closed loop”), it is even more important to tailor diabetes management to individual needs, in order to improve the outcome by customization of individual settings.

In our study, we aimed to assess the long-term effects of the treatment change to CSII and CGM in people with T1D and unsatisfactory control on MDI. After an intensive multidisciplinary and structured diabetes education, participants were offered CSII, and thereafter continuous glucose monitoring (CGM).

## 2. Experimental Section

The prospective observational single-centre cohort study included people with T1D attending a single tertiary-care medical centre at the Lucerne Cantonal Hospital, Lucerne, Switzerland. The study was approved by the ethics committee of Central Switzerland, Lucerne, Switzerland, and the study was conducted according to the Declaration of Helsinki. All included patients provided informed consent and agreement to use their anonymized data for analysis. For reporting, we adhered to the STROBE (Strengthening the Report of Observational Studies in Epidemiology) criteria for cohort studies.

The cohort comprised people with T1D who completed a course in flexible intensified insulin treatment (“functional insulin therapy” (FIT)): in addition to the regular diabetes education, a modified seminar based on Howorka’s work [6] was offered to all people with T1D. In brief, the seminars consisted of seven tutorials aiming to determine the individual fasting insulin, the bolus insulin (calculated as units of insulin per 10 g of carbohydrates), and autonomous adjustment of the insulin dose in particular settings, such as hyperglycaemia, physical activity, illness, and alcohol consumption. Groups of 10 to 12 participants with T1D and a multidisciplinary diabetes team (diabetologist, diabetes specialist nurse and dietician) elaborate and practice skills for advanced diabetes management. Thereafter, yearly updates and refreshing courses are provided. The course of the HbA1c and the cumulative incidence of hypoglycaemia and microvascular complications were compared to an observational cohort on MDI and satisfactory diabetes control. The participants of the observational cohort were also recruited after completion of the course in functional insulin therapy, and they underwent the same follow-up procedure as the intervention group.

The mean daily carbohydrate intake as well as the bolus insulin rate were calculated on the basis of annually performed “meal days” that assessed the individual consumption of carbohydrates and the corresponding bolus insulin (expressed as units of insulin per 10 g of carbohydrates). The basal insulin rate (expressed as units of insulin per 24 h) was established by performing “fasting days”, i.e., assessing plasma glucose profiles over a 24-h fast and adjusting the basal insulin dose in order to keep the glucose levels between 4 and 8 mmol/L.

From January 2006 to December 2008, participants of FIT courses were enrolled in our cohort study. Inclusion criteria were as follows: HbA1c > 7.5%, completion of FIT course, duration of T1D of at least 12 months, and reimbursement by the health insurance for CSII and CGM. Regular outpatient controls were carried out every three to six months and disease-related and treatment-related data were collected until December 2016. The insulin pumps/glucose sensors used were MiniMed® 640G, Accu-Chek® Combo, Omnipod®, FreeStyle Libre®, and Dexcom G4®, according to the choice of the participants. For recruitment to the observational cohort, inclusion criteria were as follows: HbA1c < 7.5%, completion of FIT course, duration of T1D of at least 12 months, and given informed consent.

Severe hypoglycemia was defined as an episode requiring assistance from another person (Grade III); as coma or seizures (Grade IV); or the necessity of intramuscular glucagon injection or intravenous glucose infusion, accompanied by a confirmatory blood glucose level < 2.8 mmol/L if possible. The definition of diabetic ketoacidosis (DKA) was based on an emergency hospitalization with blood glucose > 14 mmol/L, low pH < 7.3, and/or low serum bicarbonate < 15 mEq/L.

The HbA1c was measured according to DCCT/NGSP (Diabetes Control and Complications Trial/National Glycohemoglobin Standardization Program) (%) and to IFCC (International Federation of Clinical Chemistry) (mmol/mol) on Cobas C Roche/Hitachi (Roche Diagnostics GmbH, D-68305 Mannheim, Germany), with a reference range of 4.8–5.9% and 29–42 mmol/mol HbA1c, respectively. The HbA1c values before 2011 were converted to the International Federation for Clinical Chemistry standards.

Microalbuminuria screening in spot urine samples was carried out in annual intervals, and confirmed in 24 h urine samples as an albumin/creatinine ratio of 3 to 30 mg per mmol, or a urinary albumin clearance of 20–200 mcg per minute (20–300 mg per liter). Fundoscopy and grading of the retinopathy according to the protocol of the Early Treatment Diabetic Retinopathy Study (ETDRS) [7] was performed by certified ophthalmologists in annual intervals.

Statistical analyses were performed using GraphPad Prism version 8.0.1 for MacOS X, GraphPad Software, La Jolla, CA, USA. Data are expressed as means and SD, unless stated otherwise. The paired *T*-test was used to assess normally distributed data (checked by the Kolmogorov–Smirnov test). The non-parametric Mann–Whitney test was applied otherwise. Two-sided *P* values < 0.05 were considered statistically significant in the above tests.

## 3. Results

The study cohort included 41 participants with T1D (21 male/20 female), 5–49 years old (mean age, 24.2 ± 10.9 years) at the time of recruitment, with a mean follow-up of 8.9 ± 2.8 years. Enrolment is depicted in Figure 1.

Baseline characteristics of the 41 participants treated with CSII and CGM are presented in Table 1, and of the 62 participants treated with MDI in Table 2. The mean duration of diabetes at the end of the study was 15.9 years in the intervention group versus 15.5 years in the observation group (*p* = 0.14), and the frequency of severe hypoglycemia was 9.7/100 patient years (PY) versus 14.5/100 PY, respectively, *p* = 0.09.

There was a decrease of the HbA1c from baseline to 12 months in the study group (−0.84%, from 8.8 ± 1.3% to 8.0 ± 1.1%, *p* = 0.0001). Twelve months after the introduction of CGM, there was a further reduction of the HbA1c (−0.41%, to 7.6 ± 1.1%, *p* = 0.051) (Figure 2 and Figure 3). In the MDI group, there were no differences of the HbA1c from baseline to 12 months, to the end of study (from 7.31% to 7.60% and to 7.65%, respectively, *p* = 0.29 and *p* = 0.36, respectively).

The cumulative incidence of newly diagnosed diabetic microvascular complications were similar in both groups with no statistically significant difference for microalbuminuria, proliferative diabetic retinopathy, and diabetic neuropathy. There was a statistically significant decrease of the frequency of severe hypoglycemia from baseline to 12 months and to the end of the study (from 9.7 to 2.2/100 PY, *p* = 0.03). The incidence of diabetic ketoacidosis was low at study entry and did not change significantly over the entire follow-up (Table 3).

Figure 4, Figure 5 and Figure 6 show the daily insulin requirements adjusted to the body weight: there were no differences for the total daily insulin dose (*p* = 0.62), and the daily basal insulin dose (*p* = 0.41), but for the daily bolus insulin dose (*p* = 0.03). For the duration of the study, the body mass index (BMI) of the study cohort rose from 23.8 ± 4.4 kg/m^2^ to 25.6 ± 4.1 kg/m^2^ (*p* = 0.042). The total daily insulin dose was lower in the CSII group compared to the MDI group (38.2 U/day versus 43.5 U/day, *p* = 0.026). However, when adjusted to body weight, there were no differences for the total daily insulin dose (0.59 U/kg bodyweight versus 0.62 U/kg bodyweight, in the CSII versus MDI group, respectively, *p* = 0.31), and the daily basal insulin dose (0.32 U/kg versus 0.29 U/kg, *p* = 0.17), except for the daily bolus insulin dose (0.25 U/kg versus 0.33 U/kg, *p* = 0.018).

During the entire follow-up, local complications were reported as follows: kinking and/or blockage of the subcutaneous catheter in 34 (82%), consecutive high blood glucose >15 mmol/L in 25 (61%), bleeding at the infusion site in 12 (29%), and infection at the infusion site needing surgery in 2 (5%) participants.

## 4. Discussion

In this study of 41 participants with T1D and unsatisfactory diabetes control, the treatment changes to CSII resulted in a significant improvement of the HbA1c levels after 12 months, and the introduction of CGM further reduced the HbA1c levels and the frequency of severe hypoglycemia. These effects were maintained over the entire study duration of almost 9 years.

There are a number of studies describing the effects of CSII on diabetes management and outcomes [8]. A recent Danish study [9] reported a significantly greater improvement of the HbA1c in 193 CSII-treated people with T1D after four years than in 386 matched MDI-treated people with T1D (HbA1c 7.8% (62 mmol/mol) versus 8.4% (68 mmol/mol), *p* < 0.001). A retrospective trial from Sweden including 272 type 1 diabetes participants treated with CSII showed a significant and persistent reduction of the HbA1c of 0.20% (2.17 mmol/mol) over 5.5 years compared with controls on MDI [10]. The best improvement in HbA1c after initiation of CSII was observed in those participants with the highest HbA1c on MDI [11]. In our study group, the mean reduction of the HbA1c of 1.23% (from a mean baseline HbA1c of 8.8 ± 1.3%) applied equally for the whole range of baseline HbA1c. This may in part be due to the concomitant and repetitive diabetes (FIT) education.

The rate of severe hypoglycemia is roughly reduced by two-thirds by CSII compared to MDI [4,12,13]. Moreover, mild-to-moderate hypoglycaemia is also reduced by about 75%. In many quality of life-studies, CSII treatment showed an accordingly significant improvement compared with MDI [4,9,14]. In our study, there was a significant reduction of severe hypoglycaemia from baseline to the end of the study in the intervention group (from 9.7 to 2.2 /100 patient-years, corresponding to a rate ratio of 4.4) as well as in the observational group (from 14.5 to 4.8 /100 PY, rate ratio 3.0), underscoring the important effect of a regular structured and multidisciplinary diabetes education. However, given the low incidence of severe hypoglycaemia in our study, the benefits of CSII/CGM may be underestimated.

To our knowledge, data on the long-term effect of CSII on diabetic complications are missing, except for the Danish study by Rosenlund et al. [9]. They reported a significant annual reduction in the urinary albumin/creatinine ratio in CSII-treated T1D versus MDI (−10.1% versus −1.2%, *p* < 0.001). The benefit of lowering HbA1c levels is difficult to predict, because HbA1c levels before the improvement of diabetes control also contribute to the progression of diabetic complications as with diabetic retinopathy, where HbA1c levels up to eight years before add to the progression of diabetic retinopathy (“metabolic memory”) [15]. For intensive insulin treatment, however, the DCCT/EDIC (Diabetes Control and Complications Trial and Epidemiology of Diabetes Interventions and Complications study) research group yielded persistent benefits up to 18 years after its application regarding the diabetic nephropathy [16]. Over the entire duration of our study of almost nine years, diabetic microvascular complications were diagnosed for the first time in about 20% of the participants after a mean diabetes duration of more than 15 years. However, this proportion was similar in the observational cohort who had HbA1c levels <7.5%, leading to the assumption that the legacy effect [17] may be compensated for by improved diabetes control. This proportion thus may be attributed to the improvement of the HbA1c in the intervention group.

The two major disadvantages of CSII are the higher cost and non-metabolic complications. In a recent meta-analysis including 11 studies from eight countries, CSII was considered cost-effective with a mean cost effectiveness ratio of about €30.000 (US$ 40.000) per quality-adjusted life year (QALY), and 0.4 to 1.1 QALY gained [18]. Other side effects, such as significant increases in weight/BMI and DKA in CSII-treated adolescents, were reported recently [19]. However, there were no differences in our study between the CSII group and the MDI group in this regard. Non-metabolic complications of the insulin pump therapy, such as infusion set problems, pump malfunction, and infusion site complications, are common [20], and are reported by almost two-thirds of CSII-treated people with T1D, emphasizing the need of continued education and medical training.

The strengths of our study are the long-term, standardized, and uniform follow-up of almost nine years. The recording of diabetic complications in a real-life setting with the inclusion of all age groups and differently motivated participants clearly underscores the effectiveness of the insulin pump therapy/CGM in T1D [21]. However, there are limitations of our study: it was not a randomized controlled trial, and the number of the study participants is limited.

In conclusion, this study demonstrates that the initiation of CSII treatment in people with T1D combined with regular structured and multidisciplinary diabetes education results in a significant improvement of the diabetes control, which is further enhanced by the introduction of CGM. These benefits with regard to glycemic control, severe hypoglycemia, and the incidence of diabetic microvascular complications were maintained over a duration of almost nine years.

## Figures and Tables

**Figure 1 jcm-08-00394-f001:**
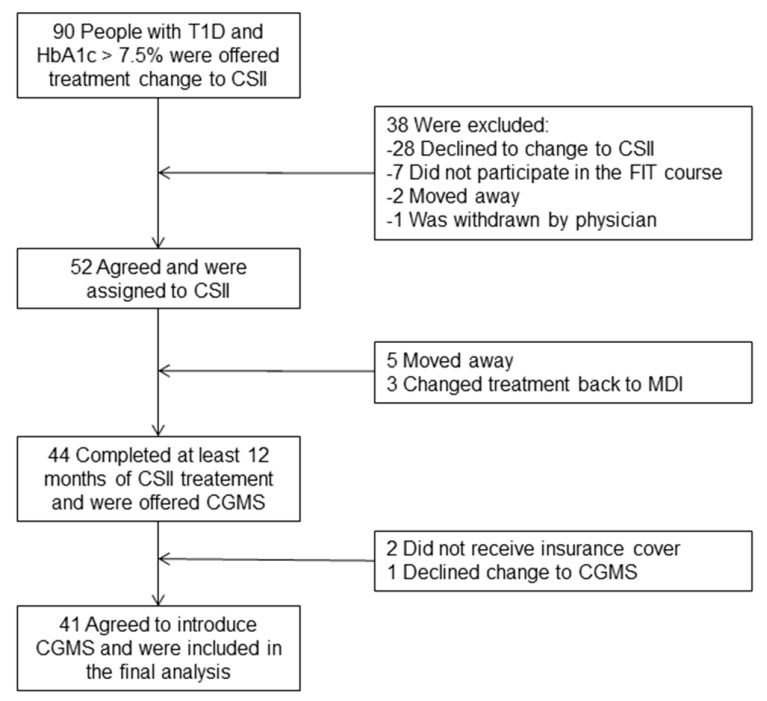
Flow diagram showing the selection of the study population. CSII: Continuous Subcutaneous Insulin Infusion; CGMS: Continuous Glucose Monitoring System; MDI: Multiple Daily Injections; FIT: Functional Insulin Therapy; T1D, type 1 diabetes.

**Figure 2 jcm-08-00394-f002:**
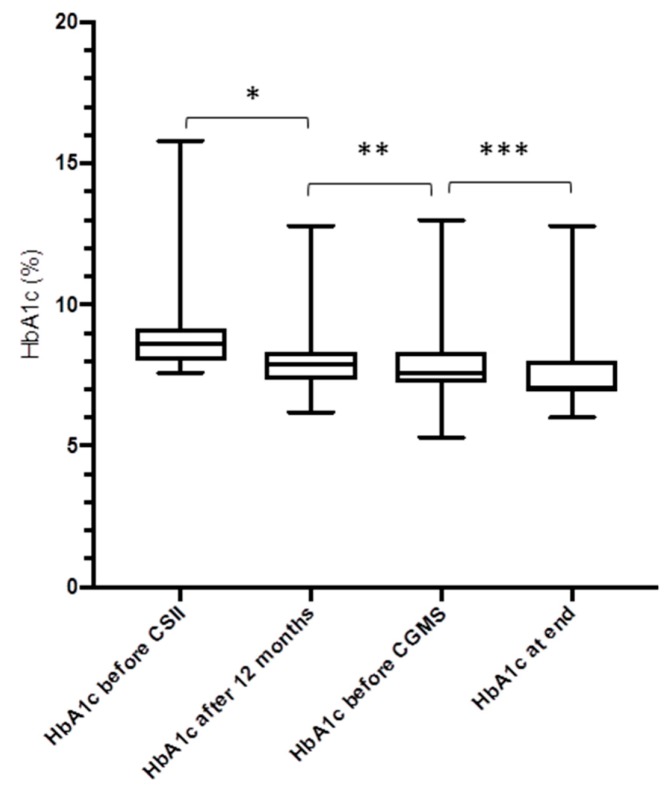
Course of the HbA1c levels in participants after treatment change to CSII and introduction of CGM (*n* = 41). * *p* = 0.003; ** *p* = 0.23; *** *p* = 0.051 (Box plots show the median and the interquartile range, and whiskers are minimum/maximum).

**Figure 3 jcm-08-00394-f003:**
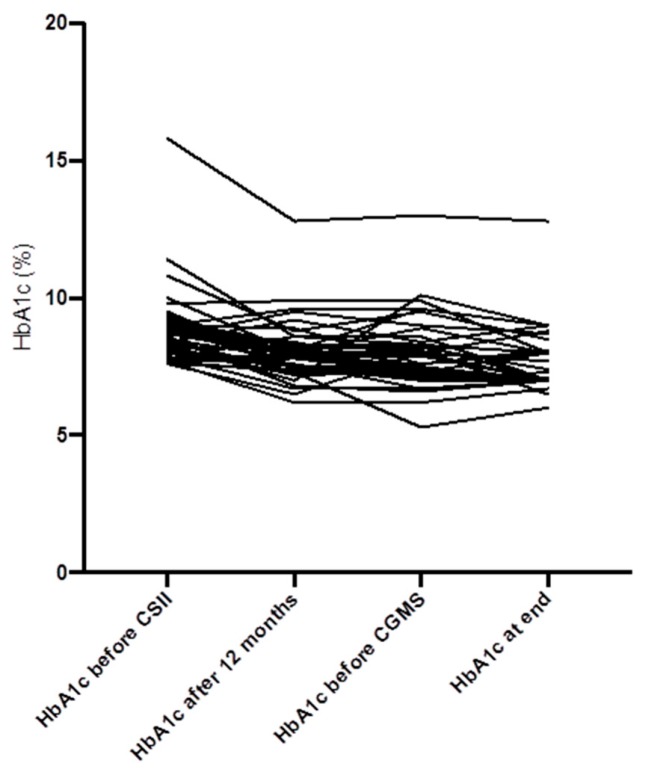
Individual course of the HbA1c levels over the entire study duration (*n* = 41).

**Figure 4 jcm-08-00394-f004:**
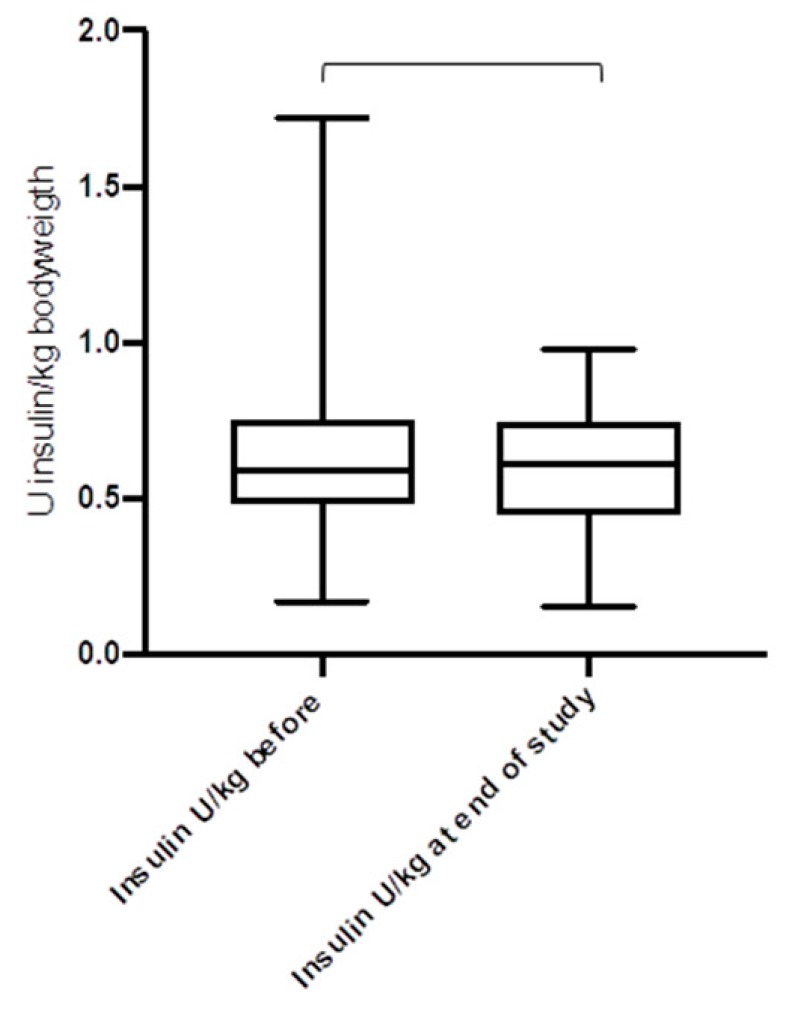
Total daily insulin dose before and after the introduction of CSII and CGM (*n* = 41) (*p* = 0.62).

**Figure 5 jcm-08-00394-f005:**
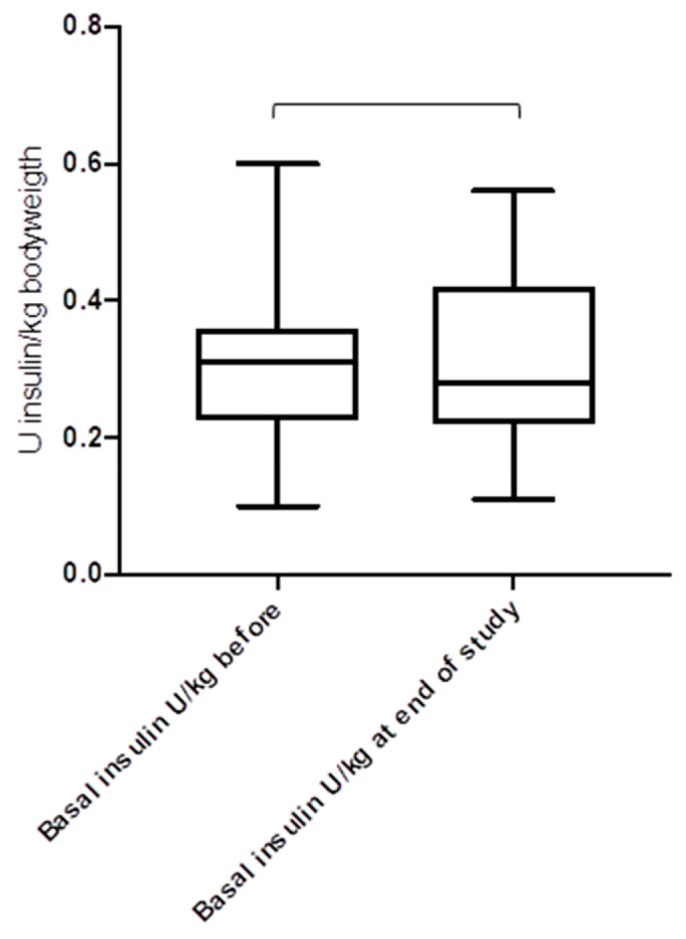
Basal insulin dose before and after the introduction of CSII and CGM (*n* = 41) (*p* = 0.41).

**Figure 6 jcm-08-00394-f006:**
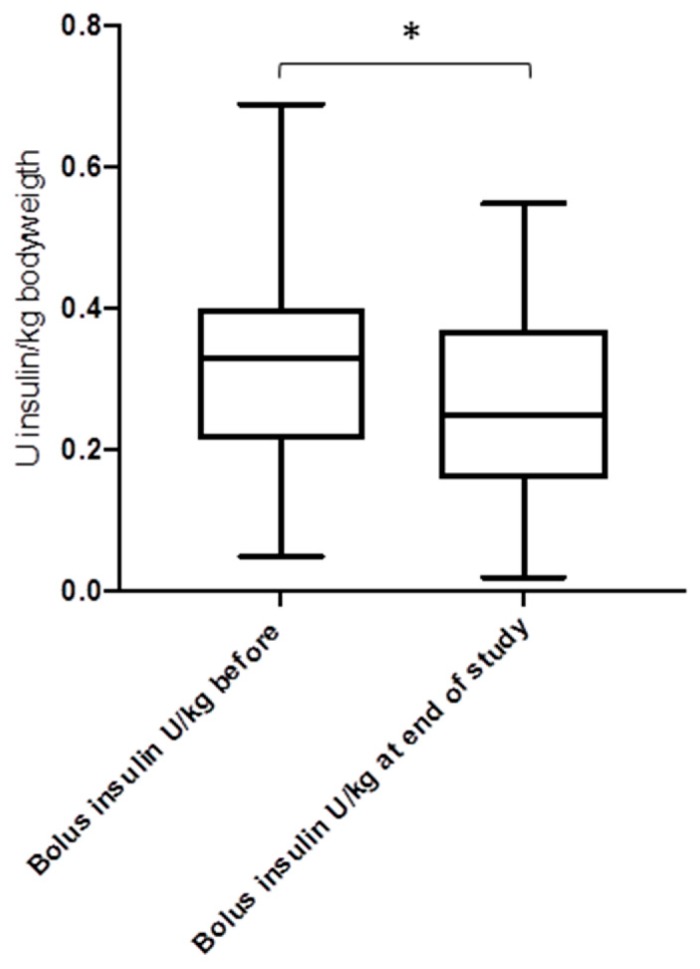
Bolus insulin dose before and after the introduction of CSII and CGM (*n* = 41) (* *p* = 0.033).

**Table 1 jcm-08-00394-t001:** Baseline characteristics of people with type 1 diabetes completing the study with treatment change to CSII and introduction of CGMS (*n* = 41).

Characteristics	*n*
Male/female (*n*)	21/20
Age at diagnosis of T1D (year)	13.7 ± 11.2
BMI (kg/m^2^)	24.0 ± 5.6
Duration of diabetes (year)	7.8 ± 9.5
Duration of follow-up (year)	8.9 ± 2.8
Duration of diabetes at study end (year)	15.9 ± 10.1
HbA1c at study entry (%)	8.85 ± 1.39
IFCC * (mmol/mol)	73 ± 8
Daily intake of carbohydrates (g)	174 ± 67
Severe hypoglycaemia (*n*/100 patient-years)	9.7
Other autoimmune disease *n* (%)	10 (24%)

Data are mean ± SD. Other autoimmune diseases are Hashimoto thyroiditis (*n* = 8), celiac disease (*n* = 3), pernicious anaemia (*n* = 2), and Addison’s disease (*n* = 1). The mean daily carbohydrate intake was calculated on the base of annually performed “meal days” that assessed the individual consumption of carbohydrates. BMI, body mass index. (* International Federation of Clinical Chemistry).

**Table 2 jcm-08-00394-t002:** Baseline characteristics of the observational cohort of people with type 1 diabetes on MDI and without CGMS (*n* = 62).

Observational Group	*n*
Male/female (*n*)	35/27
Age at diagnosis (year)	16.5 ± 12.7
BMI (kg/m^2^)	23.7 ± 3.4
Duration of diabetes (year)	8.5 ± 10.8
Duration of follow-up (year)	6.8 ± 3.2
Duration of diabetes at study end (year)	15.5 ± 13.9
HbA1c at study entry (%)	7.30 ± 0.9
IFCC (mmol/mol)	56 ± 13
Daily intake of carbohydrates (g)	216 ± 62
Severe hypoglycaemia (*n*/100 patient-years)	14.5
Other autoimmune disease *n* (%)	9 (14%)

Data are mean ± SD. Other autoimmune diseases are Hashimoto thyroiditis (*n* = 7), celiac disease (*n* = 2), and pernicious anaemia (*n* = 2).

**Table 3 jcm-08-00394-t003:** Diabetic and microvascular complications diagnosed during follow-up.

	Intervention Group (*n* = 41)	Observational Group (*n* = 62)	*P* Value
Microalbuminuria, *n* (%)	10 (24%)	13 (21%)	*p* = 0.62
Diabetic proliferative retinopathy, *n* (%)	7 (17%)	12 (19%)	*p* = 0.72
Diabetic neuropathy, *n* (%)	5 (9%)	11 (14%)	*p* = 0.44
Severe hypoglycaemia grade III/IV (*n*/100 PY)	2.2	4.8	*p* = 0.64
Ketoacidosis (*n*/100 PY)	1.8	1.9	*p* = 0.91

Hypoglycaemia and ketoacidosis are given as number per 100 patient years (PY). Comparison of groups was made with Student’s *t*-test.

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
