# Peer review of "Long-Term Effects of Initiating Continuous Subcutaneous Insulin Infusion (CSII) and Continuous Glucose Monitoring (CGM) in People with Type 1 Diabetes and Unsatisfactory Diabetes Control"

_jcm, 2019, doi:10.3390/jcm8030394_

Reviewer 1 Report

In the manuscript, Long-tern effects of initiating continuous subcutaneous insulin infusion (CSII) and continuous glucose monitoring (CGM) in people with type 1 diabetes and unsatisfactory diabetes control, Senn and colleagues have followed a cohort in there clinic with less than adequate diabetes control based on A1C who agreed to transition to CSII therapy and then to CGM therapy compared to a cohort that had adequate control and stayed on multiple daily injections (MDI). The authors have shown that in the group with less than adequate control the transition to CSII and CGM helped them to lower their A1C.

However, there are some concerns raised with regard to the control cohort selection, data analysis and conclusions.

The authors have included a very nice diagram and description of the criteria for the selection of the cohort that did not have adequate control; however, the authors did not include any description of the inclusion or exclusion criteria for the 62 members of the control cohort. This data must be included.

The authors included 2 tables for the descriptive statistics of the cohort, but in doing this, there was no statistical comparison of the 2 groups. This must be added. In looking at the 2 tables, it appears that the control group was older at diagnosis. Were they older at the time of the study? The age at the start of the trial was not included for either group and may be of use in explaining why the A1C was different (i.e. the CSII were mostly teenagers and the MDI group was adults?). Does the difference in A1C relate to age?

In Figure 3, it appears that one participant was any outlier with an initial A1C of greater than 15%. Was any analysis of outliners done? If this participant was excluded, were any of the results affected? This needs further analysis.

In Figures 4 and 5, and asterisks is included for the p-value, but the p-value is not significant. The p-value may be mentioned but the asterisks should be removed.

The authors found that the total daily insulin dose and basal dose per kg did not change, but the bolus dose was decreased. How were the bolus insulin calculated prior to the initiation of pump therapy? If it was based on patient recall, this may be a source of bias. The these methods need to be described. Also there results were not reported in the MDI group which would be a useful comparison.

The authors make the statement that CSII and CGM lowered the frequency of hypoglycemic events, but there was not a report of the statistic to back this finding. It appears from Table 3 that hypoglycemic events decreased in both the CSII and the MDI group. It would be useful to compare the baseline incidents to the follow-up incidents. In the discussion the authors quote other studies about the reduction in hypoglycemia with CSII, but do not include the numbers for the present study.

The authors mention in the discussion the effects of CSII on complications, but in their analysis, there was no difference in complication rates between the CSII and the MDI groups. This warrants a discussion in this section.

Minor edit: Line 167, too many spaces between years and than

Author Response

Response to Reviewer 1

Dear Reviewer 1

Thank you very much for the review of our manuscript and the highly appreciated suggestions to improve it ! The suggested changes were implemented and highlighted in the revised manuscript. We would like to give the following answers to the comments.

The authors have included a very nice diagram and description of the criteria for the selection of the cohort that did not have adequate control; however, the authors did not include any description of the inclusion or exclusion criteria for the 62 members of the control cohort. This data must be included.

Implemented as : The participants of the observational cohort were also recruited after completion of the course in functional insulin therapy, and they underwent the same follow-up procedure as the intervention group.

For recruitement to the observational cohort inclusion criteria were as follows: HbA1c < 7.5%, completion of FIT course, duration of T1D of at least 12 months, and given informed consent.  

The authors included 2 tables for the descriptive statistics of the cohort, but in doing this, there was no statistical comparison of the 2 groups. This must be added. In looking at the 2 tables, it appears that the control group was older at diagnosis. Were they older at the time of the study? The age at the start of the trial was not included for either group and may be of use in explaining why the A1C was different (i.e. the CSII were mostly teenagers and the MDI group was adults?). Does the difference in A1C relate to age?

The statistical comparison of the intervention and observational group will be given in the result section. There was a statistically significant difference at the age of diagnosis, however, the duration of diabetes (which is crucial for complications) at study end was not different (cf. table below). Because this was not a double-blind and randomized trial with an intervention and a control group, we suggest to mention these p-values in the Result Section, but to leave the tables separated. There was no statistically significant correlation of A1C and age in either group.

In Figure 3, it appears that one participant was any outlier with an initial A1C of greater than 15%. Was any analysis of outliners done? If this participant was excluded, were any of the results affected? This needs further analysis.

Considering the absolute change in A1C, there was no change in the outcome if separate analysis of the outliers (e.g. those with A1C >10%) were done. Further, there were no correlations between the A1C at start of the study and the decrease in the A1C levels.  

In Figures 4 and 5, and asterisks is included for the p-value, but the p-value is not significant. The p-value may be mentioned but the asterisks should be removed.

Implemented as proposed.

The authors found that the total daily insulin dose and basal dose per kg did not change, but the bolus dose was decreased. How were the bolus insulin calculated prior to the initiation of pump therapy? If it was based on patient recall, this may be a source of bias. The these methods need to be described. Also there results were not reported in the MDI group which would be a useful comparison.

Implementation in the Experimental section:

The mean daily carbohydrate intake as well as the bolus insulin rate were calculated on the base of annually performed “meal days” with assessing the individual consumption of carbohydrates and the corresponding bolus insulin (expressed as Units of insulin per 10 g of carbohydrates). The basal insulin rate (expressed as Units of insulin per 24 hours) was established by performing “fasting days”, i.e. assessing plasma glucose profiles over a 24 hour fast and adjusting the basal insulin dose in order to keep the  glucose between 4 and 8 mmol/l.

Both groups (independent whether they changed to CSII/CGM or not) had annually updates/refreshing courses of the flexible insulin therapy (FIT). In these settings there were “fasting days” to assess the basal insulin need, and “meal days” to assess the bolus insulin need on the base of the consumption of carbohydrates. For the individual intake of carbohydrates the mean of these yearly measurements were used.

Implemented in the Result section:

The total daily insulin dose was lower in the CSII group compared to the MDI group (38.2 U/d vs. 43.5 U/d, p=0.026). However, when adjusted to the body weight there were no differences for the total daily insulin dose (0.59 U/kg bodyweight vs. 0.62 U/kg bodyweight, in the CSII vs. MDI group, respectively, p=0.31), and the daily basal insulin dose (0.32 U/kg vs. 0.29 U/kg, p=0.17), except for the daily bolus insulin dose (0.25 U/kg vs. 0.33 U/kg, p=0.018).

The authors make the statement that CSII and CGM lowered the frequency of hypoglycemic events, but there was not a report of the statistic to back this finding. It appears from Table 3 that hypoglycemic events decreased in both the CSII and the MDI group. It would be useful to compare the baseline incidents to the follow-up incidents. In the discussion the authors quote other studies about the reduction in hypoglycemia with CSII, but do not include the numbers for the present study.

There was a significant reduction of severe hypoglycaemia in both group (also pointing to the effect of the FIT courses providing  regular structured and multidisciplinary diabetes education) : in the intervention group from 9.7 to 2.2 /100 patient-years (corresponding to rate ratio of 4.4), and in the observation group from 14.5 to 4.8 /100 PY (rate ratio 3.0), p=0.64. These numbers numbers were inserted in the revised text.

Insertion in the revised text: In our study there was a significant reduction of severe hypoglycaemia from baseline to the end of the study in the intervention group (from 9.7 to 2.2 /100 patient-years, corresponding to rate ratio of 4.4) as well as in the observational group (from 14.5 to 4.8 /100 PY, rate ratio 3.0), underscoring the important effect of a regular structured and multidisciplinary diabetes education.

The authors mention in the discussion the effects of CSII on complications, but in their analysis, there was no difference in complication rates between the CSII and the MDI groups. This warrants a discussion in this section.

The results shown on table 3. are commented in the Discussion section : The frequency of diabetic microvascular complications at study end were similar in the CSII and in the MDI group what may attributed to the improvement of the HbA1c in the intervention group.

Minor edit: Line 167, too many spaces between years and than

This has been corrected.

Reviewer 2 Report

Major points

Abstract, line 25, instead of stating “ declined significantly  “, please state by how much (and keep p=0.03)

Introduction, line 34, please change to read “In patients with type 1 diabetes…”

Introduction line 51-52 – Please state how much more beneficial

Introduction line 52-53 – please explain this statement, in my view, the artificial pancreas should diminish the need to tailor invidualized therapy.

Introduction lines 57-58 Please put this sentence after you describe the experimental cohort (lines 66-74), and please clarify, is it contemporaneously observed. Also please clarify if the comparator cohort not using CGM had a similar amount of visits with physicians and other members of their care team.

Introduction, lines 79-80, what determined which CGM system was used?

Line 82, How was coma assessed (did you use Glasgow coma scale)? What was used to define coma? Are individuals with altered mental status, but responsive to noxious stimuli considered to have coma?

 Line 82, glucagon may also be given subcutaneously, do you mean to include that as well as IM glucagon as an adverse outcome?

Line 84-85, Please confirm the definition of DKA, serum glucose above 14 mmol/L is not a requirement. Were ketones measured or anion gap?

Tables 1 and 2 should be combined, and each data point should be compared statistically to generate a p-value between the 2 groups.

Tables 1 and 2, please indicate in the methods section how daily intake of dietary carbohydrates was assessed. In addition, how do you control for the possibility that dietary carbohydrate intake may change over the duration of the trial?

Line 132 please remove the words “statistically significant”, this is an interpretation and belongs in the discussion, not the results section. Similarly, please remove “comparable” from line 108 and reword the sentence to just tell the results.

Line 134 please say “reduction” instead of “improvement”, and please re-word for clarity as to what time point you are referring to.

Line 167-168 please change “diabetes people” to “people with T1D”

Lines 183-185, please re-word the sentence “The  benefit of lowering HbA1c levels…” for clarity.

Lines 196-198 please re-word for clarity.

Minor points

Introduction line 36, please change “get” to “attain” or some other synonym for style.

Line 40 change “or” to “through”

Line 48 define HbA1c

Line 49 change “meta-analysis” to “meta-analyses

Line 56-57 might be best to use only CGMS (line 41-42) or CGM, no need for both abbreviations.

Line 80 please change “Grad” to “Grade” in both instances.

Table 1, line 114 is confusing and needs header text.  Male/Female is not the header text and should not be bold.

Author Response

Response to Reviewer 2

Dear Reviewer 2

Thank you very much for the review of our manuscript and the highly appreciated suggestions to improve it !

All the major points below were implemented and highlighted in the revised manuscript. 

Major points

Abstract, line 25, instead of stating “ declined significantly  “, please state by how much (and keep p=0.03)

Implemented as proposed.

Introduction, line 34, please change to read “In patients with type 1 diabetes…”

Implemented as proposed.

Introduction line 51-52 – Please state how much more beneficial

Implemented as :with a range of the mean difference of the HbA1c before/after studies of 0.21 to 0.72%...

Introduction line 52-53 – please explain this statement, in my view, the artificial pancreas should diminish the need to tailor invidualized therapy.

Implemented as :  in order to improve the outcome by customization of individual settings.

Introduction lines 57-58 Please put this sentence after you describe the experimental cohort (lines 66-74), and please clarify, is it contemporaneously observed. Also please clarify if the comparator cohort not using CGM had a similar amount of visits with physicians and other members of their care team.

Implemented as proposed : The course of the HbA1c and the cumulative incidence of hypoglycaemia and microvascular complications were compared to an observational cohort on MDI and satisfactory diabetes control The participants of the observational cohort were also recruited after completion of the course in functional insulin therapy, and they underwent the same follow-up procedure as the intervention group.

Introduction, lines 79-80, what determined which CGM system was used?

Implemented as : The insulin pumps/glucose sensors used were MiniMed® 640G, Accu-Chek® Combo, Omnipod®, FreeStyle Libre®, and Dexcom G4®,according to the choice of the participants. 

Line 82, How was coma assessed (did you use Glasgow coma scale)? What was used to define coma? Are individuals with altered mental status, but responsive to noxious stimuli considered to have coma?

A modified version of the American Diabetes Association (ADA) and Endocrine Society recommendations for the definition and classification of hypoglycaemic coma were applied, i.e. a) asymptomatic hypoglycaemia (blood glucose<3.9mmol/l without symptoms, grade 1) ; b) symptomatic hypoglycaemia (with typical symptoms of hypoglycaemia, grade 2) ; c) severe hypoglycaemia (needing assistance to administer carbohydrates or glucagon, grade III) ; d) hypoglycaemic coma (loss of consciousness or seizures corresponding to Glasgow coma scale <9, grade IV). 

 Line 82, glucagon may also be given subcutaneously, do you mean to include that as well as IM glucagon as an adverse outcome?

The administration of glucagon was considered as a point for the severity of hypoglycaemia (i.e. grade III and IV)

Line 84-85, Please confirm the definition of DKA, serum glucose above 14 mmol/L is not a requirement. Were ketones measured or anion gap?

In the routinely performed blood gas analysis the anion gap (and Bicarbonate levels) were measured, and blood samples for the measurement of ketones were stored (in case of unclear diagnosis).

Tables 1 and 2 should be combined, and each data point should be compared statistically to generate a p-value between the 2 groups.

Because this was not a double-blind and randomized trial with an intervention and a control group, we suggest to mention the p-values in the Result Section, but to leave the tables separated. 

Tables 1 and 2, please indicate in the methods section how daily intake of dietary carbohydrates was assessed. In addition, how do you control for the possibility that dietary carbohydrate intake may change over the duration of the trial?

Implementation in the text: The mean daily carbohydrate intake was calculated on the base of annually performed “meal days” with assessing the individual consumption of carbohydrates.

Both groups (independent whether they changed to CSII/CGM or not) had annually updates/refreshing courses of the flexible insulin therapy (FIT). In these settings there were “fasting days” to assess the basal insulin need, and “meal days” to assess the bolus insulin need on the base of the consumption of carbohydrates. For the individual intake of carbohydrates the mean of these yearly measurements were used.

Line 132 please remove the words “statistically significant”, this is an interpretation and belongs in the discussion, not the results section. Similarly, please remove “comparable” from line 108 and reword the sentence to just tell the results.

Implemented as proposed.

Line 134 please say “reduction” instead of “improvement”, and please re-word for clarity as to what time point you are referring to.

Implemented as proposed.

Line 167-168 please change “diabetes people” to “people with T1D”

Implemented as proposed.

Lines 183-185, please re-word the sentence “The  benefit of lowering HbA1c levels…” for clarity.

Implemented as : The benefit of lowering HbA1c levels is difficult to predict, because also HbA1c levels before the improvement of diabetes control contribute to the progression of diabetic complications as with diabetic retinopathy, where HbA1c levels up to 8 years before add to the progression of diabetic retinopathy (“metabolic memory”) [15]

Lines 196-198 please re-word for clarity.

Implemented as : Other side effects like  significant increase in weight/BMI and DKA in CSII treated adolescents were reported recently [19].

All the minor points below were implemented as suggested, and highlighted in the revised manuscript.   

Minor points

Introduction line 36, please change “get” to “attain” or some other synonym for style.

Line 40 change “or” to “through”

Line 48 define HbA1c

Line 49 change “meta-analysis” to “meta-analyses”

Line 56-57 might be best to use only CGMS (line 41-42) or CGM, no need for both abbreviations.

Line 80 please change “Grad” to “Grade” in both instances.

Table 1, line 114 is confusing and needs header text.  Male/Female is not the header text and should not be bold.

Round  2

Reviewer 1 Report

 The authors Senn and colleagues in their manuscript, "Long-term effects of initiating continuous subcutaneous insulin infusion (CSII) and continuous glucose monitoring (CGM) in people with type 1 diabetes and unsatisfactory diabetes control," have addressed some of my concerns raised in my review. The main concern still lies in the age of the two cohorts. In the response the authors state that the averages age of the control group was 26 years. In the manuscript is states that it is 16 years. The p-value in the response would suggest that there is a difference in the age of the 2 cohorts which is not mentioned in the discussion. This is a very important finding as many studies including T1D Exchange and others show that the A1c is lower in adults that an in teens. The improvement in A1c may not be related to the use of technology, but rather to maturation of the cohort (they were followed for 9 years).  This is still my greatest concern with the manuscript.

 Also in line 78 there are too many spaces

In line 143, "twelve" is misspelled

Author Response

Dear Reviewer 1,

Thank you so much for your comment that pointed to the difference in "Age at diagnosis":

The main concern still lies in the age of the two cohorts. In the response the authors state that the averages age of the control group was 26 years. In the manuscript is states that it is 16 years. The p-value in the response would suggest that there is a difference in the age of the 2 cohorts which is not mentioned in the discussion. This is a very important finding as many studies including T1D Exchange and others show that the A1c is lower in adults that an in teens. The improvement in A1c may not be related to the use of technology, but rather to maturation of the cohort (they were followed for 9 years).  This is still my greatest concern with the manuscript.

We deeply regret our mistake in providing the provisional table included in our first response to your comments. Actually, the correct number for the age of the MDI group is the one of Table 2 in the manuscript indicating 16.5+12.7 years (the mistake went unnoticed because the reply and the table were done by two different authors). Nevertheless, there remains a difference with a p-value of 0.034. However, the participants in both groups were mainly in their adolescence, and the study was designed to demonstrate the effects of the CSII/CGM in the Intervention Group. The observational cohort was primarely recruited to compare the outcome in relation to diabetic complications like hypoglycaemia and ketoacidosis, and to microvascular complications like microalbuminuria, retinopathy and neuropathy. Because of the longer follow-up duration in the intervention group, the duration of diabetes at study end was very similar in both groups (15.9.vs. 15.5 years), and, therefore, allowed a clear statement regarding this issue, and, further, an affirmative answer to the question whether there are long-term benefits of the CGM/CSII (cf. Pickup JC).

We very much appreciate your help in improving our manuscript!

(Pickup JC. Is insulin pump therapy effective in Type 1 diabetes? Diabet Med. 2018 Aug 11. doi: 10.1111/dme.13793).

Also in line 78 there are too many spaces

In line 143, "twelve" is misspelled

Both mistakes were corrected.